# Identification of Cancer Cells in the Human Body by Anti-Telomerase Peptide Antibody: Towards the Isolation of Circulating Tumor Cells

**DOI:** 10.3390/ijms232112872

**Published:** 2022-10-25

**Authors:** Olga Karpov, Meir Lahav, Ofir Wolach, Pia Raanani, Dan Peer, Tal Kaplan, Orit Uziel

**Affiliations:** 1Laboratory of Precision Nano-Medicine, School of Molecular Cell Biology and Biotechnology, and Aladar Fleischman Faculty of Engineering, Center for Nanoscience and Nanotechnology, Cancer Biology Research Center, Tel Aviv University, Tel Aviv 69978, Israel; 2The Felsenstein Medical Research Center, Rabin Medical Center, Petah Tikva 49100, Israel; 3Sackler School of Medicine, Tel-Aviv University, Tel Aviv 69978, Israel; 4Institute of Hematology, Davidoff Cancer Center, Rabin Medical Center, Petah Tikva 49100, Israel; 5BioView Ltd., Rehovot 7670203, Israel

**Keywords:** telomerase, circulating tumor cells, pan cancer marker

## Abstract

Early detection of tumor cells by identifying universal Tumor Associated Antigens (TAA) can drastically change our diagnostic, theranostic and therapeutic possibilities to cure cancer. Human Telomerase Reverse Transcriptase (hTERT), a hallmark of cancer, could act as an optimal TAA candidate. Here we report about the development of a monoclonal antibody against hTERT peptide (α-hTERT mAb) presented on the surface of cancer cells and its possible applications as a pan-cancer marker. Liquid biopsies, an innovative tool in precision oncology, comprising the noninvasive analysis of circulating tumor-derived material to counteract limitations associated with tissue biopsies. Within the tumor circulome, the US Food and Drug Administration already approved the use of circulating tumor cells (CTCs) as valid liquid biopsies. However, currently CTCs are being trapped using antibodies against specific cancer types, with anti EpCAM as the most common antibody, directed mainly against solid tumors. Moreover, the precision medicine approach is based on specific cancer type directed antibodies. Our novel mAb against the hTERT 16-mer peptide, corresponding to amino acids 611–626, is capable of detecting various types of cancer cells both in vitro and ex vivo from tumors of patients with either hematological or solid tumors. This antibody does not bind to normal lymphocytes cells. Cleavage of our antibody to F(ab’)2 fragments increased its binding specificity to the tested cancer cells. Future studies may point to the use of this antibody in the procedure of capturing CTCs.

## 1. Introduction

We are in the midst of major advances in understanding cancer biology and in development of rationalized biological drugs against specific targets in cancer cells [1]. However, the area of early diagnosis still lags behind. Although the ability to study solid cancers by noninvasive sampling of blood (or other body fluids), namely liquid biopsies, sounds promising in cancer diagnostics, there are still significant limitations hindering their clinical usage. Liquid biopsies are manifested by two major strategies: cell-free circulating tumor DNA (ctDNA) and circulating tumor cells (CTC) [2]. Derived from tumor deposits or lysed CTCs, the relatively simple isolation of ctDNA has been shown to exhibit clinical potential for diagnostic purposes. However, despite relative simplicity in its isolation, the small amount of ctDNA is highly diluted within the larger amount of ctDNA originating from normal cells (comprising 0.7% of the total DNA), thus restricting its potential use. The development of Next Generation Sequencing techniques may serve as a possible solution for this problem. In addition, an optimal analysis and follow up of ctDNA requires a prior elucidation of the tumor DNA sequence [3]. Utilization of CTCs as liquid biopsies, on the other hand, does not require any statistical manipulations and calculations. However, this method has its unique intrinsic limitations. CTCs comprise about 10–1000 cells among 10^9^ Red Blood Cells (RBCs) and 5000–10,000 leukocytes. Therefore, their isolation should be very specific [4].

The most common strategy for capturing CTCs is based on the expression of specific markers on their membrane. By using the EpCAM and CK markers, the commercial technology of CellSearch enables the isolation of epithelial derived tumor cells by magnetic based antibodies [5,6,7]. Although reproducible, this technique suffers from a relatively low recovery yield. Other EpCAM based isolation technologies (“CTC Chip” platforms) are not suitable for high throughput assays [8,9]. The use of the EpCAM marker, however, restricts the ability to detect CTCs to epithelial cells only.

In order to overcome these drawbacks, new isolation strategies were being developed based on differences in their properties such as size, adherence, density, electric charge of tumor versus normal cells [10,11,12,13,14,15,16,17]. The disadvantage of the size-based approach stems from the variation in cell size and the necessity to use large volumes of cells. A recent technology is based on the depletion of normal leukocytes from the blood sample [2]. This approach requires a massive depletion of leukocytes and is currently based on sophisticated microfluidic technologies, which are not routinely available [10]. Another novel and efficient technique-CTC-iChip-uses size separation followed by a flow through microfluidic channel and separation according to their inertial focusing. This technique also requires specific equipment that is not commonly used [8].

Currently, there is no validated universal marker that enables a successful identification of large variety of CTCs as an early diagnosis strategy. Here we describe our approach using telomerase peptide presented on the membrane of all cancer cells for the development of an anti-telomerase antibody-based tool for pan cancer cells CTC isolation. Telomerase is the hallmark of the cancer cell. It is highly expressed in almost all types of cancer cells and is repressed in most somatic cells [18]. A panel of hTERT peptides expressed at the outer membrane of cancer cells capable to induce cytotoxic T-lymphocytes (CTLs) have been identified [19,20]. Previously, hTERT peptides were used to construct a vaccine against cancer cells of various types, such as non-small cell lung cancer [21]. Studies using telomerase peptides as a vaccination approach showed that it may be used as an attractive target for novel immunotherapies against cancer, by CTLs that can recognize peptides derived from TERT and eliminate TERT-positive tumor cells in murine and humans [22]. Herein we used one of these peptides termed GV1001, a 16 amino acid peptide (EARPALLTSRLRFIPK) [23] to generate an antibody that identifies cancer cells expressing telomerase. Our antibody is specific to telomerase positive cells, both in vitro and ex vivo. It efficiently identifies CTCs derived from the peripheral blood samples of patients with various malignancies, thus establishing the use of anti-telomerase-based technology as a tool for pan-cancer CTCs isolation.

## 2. Results

### 2.1. The α-hTERT Monoclonal Antibody Binds to Various Cancer Cells but Not to Fibroblasts

The specificity of the newly synthesized α-hTERT mAb (designed as α-hTERT) was tested by assessing its ability to bind specifically the 16-amino acid telomerase peptide of which it was designed against (not shown). The binding of the newly synthesized α-hTERT mAb to various cancer cell lines and fibroblasts (non-cancer cells) was tested using a flow cytometry. The following cells were assayed: Jurkat (T cell leukemia), MCF-7 (breast adenocarcinoma), U266 (multiple myeloma), MEL (melanoma), JeKo-1 (mantle cell lymphoma), A549 (lung carcinoma). Fibroblast cells served as a negative control. The ability of the antibody to bind these cells is presented in Figure 1A, as percentage of a-hTERT positive cells. α-hTERT mAb binds to various cancer cell lines at different percentages depending on affinity and expression of the target. Fibroblasts, which do not express hTERT, did not bind the antibody. Similarly, immunostaining of two carcinomas of the colon, SW480 and HCT116 were successfully labelled with the α-hTERT mAb while PBMCs from a healthy donor did not (Figure 1B).

### 2.2. The α-hTERT mAb Binds to Cancer Cells Isolated from CLL Patients

The binding of α-hTERT mAb to cancer cells isolated from the peripheral blood of patients with CLL (ex vivo) was tested as described in Materials and Methods. A total of 57 blood samples at different stages of the disease (4–96% CLL cells according to their clinical data), including primary diagnosed and treated, were tested. The binding to 4 representative blood samples of CLL and 4 solid tumors samples of different origin are shown in Figure 2 (graphs) and Figure 3 (dot plots+graphs), using the isotype IgG1 as a control. Representative dot plots are shown in Figure 3: Figure 3A depicts that most of the α-hTERT positive PBMCs (72%) are CD19+/CD5+. Figure 3B demonstrates that the α-hTERT mAb binds CLL cells gated as CD5+/CD19+. Figure 3C,D depict that the α-hTERT mAb binds ex vivo to CD19+/CD5+ cells from the newly diagnosed patient and similarly diagnosed patient after treatment against CLL, respectively. Figure 4 shows that the percentage of the α-hTERT mAb positive cells detected by the flow cytometry in Tel-Aviv university laboratory is very similar to the percentage of CLL (CD19+/CD5+) cells identified by the flow cytometry unit at the Beilinson hospital.

The lack of α-hTERT mAb’s binding to normal lymphocytes is depicted in Figure 5 representing a negative (control) population. As shown, there was no binding of α-hTERT mAb to the lymphocytes obtained from healthy volunteers. These results demonstrate the specificity of the binding of the α-hTERT mAb to malignant lymphocytes.

### 2.3. Ex-Vivo Binding of α-hTERT mAb to Cancer Cells Isolated from Patients with Solid Tumors

5 ovarian tumors and 9 ascites samples from ovary cancer patients were analyzed. Appendix A Appendix A demonstrates the binding of the α-hTERT mAb to cells from patients with ovarian carcinoma. The main population was incubated with α-hTERT mAb or the isotype control, the population of interest named FL1+ representing 4% of the main population is surrounded by a rectangle. A notable shift of this population is obtained following staining with α-hTERT mAb. Importantly, when stained with an additional marker, CCR5, α-hTERT positive cells were also CCR5 positive as shown in Appendix A Appendix A. Isotype IgG1 and control AF647 IgG served as nonspecific binding control.

The binding of α-hTERT mAb to PBMCs of a patient with Merkel carcinoma is presented in Appendix A Appendix A. Although with no specific markers for Merkel carcinoma we are unable to confirm that the α-hTERT mAb specifically binds to cancer cells, a clear shift of cells subpopulation is detected following the incubation with α-hTERT mAb and not with isotype IgG.

The *Ex vivo* binding of the α-hTERT mAb to cells obtained from three patients with breast cancer is shown in Appendix A Appendix A. A subset of cells is clearly positive to FITC-α-hTERT compared to that of the isotype control.

In addition, we were able to detect the binding of the α-hTERT mAb to a small subset of cells from two patients with lung carcinomas as presented by dot plots on Appendix A Appendix A.

### 2.4. F(ab’)2 Fragment of the α-hTERT mAb: Purification and Ex-Vivo Binding to CLL Cells

In order to optimize the binding specificity of the α-hTERT mAb to the TERT antigen, the F(ab’)2 fragment was cleaved off using ficin. The F(ab’)2 fragment was detected by gel electrophoresis (SDS-PAGE), and its binding to CLL cells was analyzed by flow cytometry. The results of the binding assays of the F(ab’)2 fragment obtained from two different preparations are demonstrated in Figure 6. As clearly shown, the F(ab’)2 fragment presented a high ex-vivo binding affinity to the PBMCs of patients with CLL as compared to that of the intact mAb or to that of the isotype IgG1 control.

### 2.5. Sequencing of the Anti-hTERT Monoclonal Antibody

The amino acid sequences of the proteins comprising the light chain of the anti-hTERT mAb was assessed by Edman degradation, and the sequencing of the heavy chain was performed using an in-gel protein digestion followed by liquid chromatography-mass spectrophotometry (LC-MS) procedures, as described in Materials and Methods. The light chain (Appendix A Appendix A) and heavy chain (Appendix A Appendix A) amino acids sequences of α-hTERT, designated herein as SEQ ID #1 and SEQ ID NO#2, respectively, are:
SEQ ID #15’-DVVMTQTPLTLSVTIGQPASISCKSSQNLLYSDGKTYLNWLLQRPGQSPKRLIYLVSKVDSGVPDRFTGSGSGTDFTLKISRVEAEDLGVYFCWQGTHLPYTFGGGTKLEIK-3’SEQ ID #25’VQLQQSGAELVRPGASVTLSCKASGYIFTDYEKHWVKQTPVHGLEWIGAIDPESGSTVYNQRFKGKATLTADKSSGTAYMELRSLTSEDSAVYFCFLLRLFAYWGQGTLVTVSA -3’

The nucleic acid sequencing was performed using the degenerate primer-based sequencing technique described in Materials and Methods. The light chain and heavy chain nucleic acids sequences of α-hTERT designated herein as SEQ ID #3 and SEQ ID #4, respectively, are:
SEQ ID #35’-GATGTTGTGATGACCCAGACTCCACTCACTTTGTCGGTGACCATTGGACAACCAGCCTCCATCTCTTGCAAGTCAAGTCAGAACCTCTTATATAGTGATGGAAAGACATATTTGAATTGGTTGTTACAGAGGCCAGGCCAGTCTCCAAAGCGCCTAATCTATCTGGTGTCTAAAGTGGACTCTGGAGTCCCTGACAGGTTCACTGGCAGTGGATCAGGGACAGATTTCACACTGAAAATCAGCAGAGTGGAGGCTGAGGATTTGGGAGTTTATTTTTGCTGGCAAGGTACTCATCTTCCGTACACGTTCGGAGGGGGGACCAAGCTGGAAATAAAACGGGCTGATGCTGCACCAACTGTATCCATCTTCCCACCATCCAGTGAGCAGTTAACATCTGGAGGTGCCTCAGTCGTGTGCTTCTTGAACAACTTCTACCCCAGAGACATCAATGTCAAGTGGAAGATTGATGGCAGTGAACGACAAAATGGTGTCCTGAACAGTTGGACTGATCAGGACAGCAAAGACAGCACCTACAGCATGAGCAGCACCCTCACATTGACCAAGGACGAGTATGAACGACATAACAGCTATACCTGTGAGGCCACTCACAAGACATCAACTTCACCCATCGTCAAGAGCTTCAACAGGAATGAGTGTTAA-3’SEQ ID #45’-caggtgCAGCTGCAGCAGTCTGGGGCTGAACTGGTGAGGCCTGGGGCTTCAGTGACGCTGTCCTGCAAGGCTTCGGGCTACATATTTACTGATTATGAAAAGCACTGGGTGAAGCAGACACCTGTGCATGGCCTGGAGTGGATTGGAGCTATTGATCCTGAAAGTGGTAGTACTGTCTACAATCAGAGATTCAAGGGCAAGGCCACACTGACTGCAGACAAATCTTCCGGCACAGCCTACATGGAACTCCGCAGCCTGACATCTGAGGATTCTGCCGTCTATTTCTGCTTTTTACTACGGCTATTTGCTTACTGGGGCCAAGGGACTCTGGTCACTGTCTCTGCAGCC-3’

The variable region of the light (VL) chain comprises the nucleotide sequence of SEQ ID #6:
SEQ ID #65’-GATGTTGTGATGACCCAGACTCCACTCACTTTGTCGGTGACCATTGGACAACCAGCCTCCATCTCTTGCAAGTCAAGTCAGAACCTCTTATATAGTGATGGAAAGACATATTTGAATTGGTTGTTACAGAGGCCAGGCCAGTCTCCAAAGCGCCTAATCTATCTGGTGTCTAAAGTGGACTCTGGAGTCCCTGACAGGTTCACTGGCAGTGGATCAGGGACAGATTTCACACTGAAAATCAGCAGAGTGGAGGCTGAGGATTTGGGAGTTTATTTTTGCTGGCAAGGTACTCATCTTCCGTACACGTTCGGAGGGGGGACCAAGCTGGAAATAAAACGGGCTGATGCTGCACCAACTGTATCCATCTTCCCACCATCCAGTGAGCAGTTAACATCTGGAGGTGCCTCAGTCGTGTGCTTCTTGAACAACTTC-3’

## 3. Discussion

Many signal transduction pathways which mediate malignant transformation have been deciphered. Accordingly, new efficient anti-cancer rationalized drugs were developed [1]. However, the diagnosis area still lags behind the therapeutic one and as such presents an unmet need [24].

Recently, a novel technology has been developed for this purpose which is based on liquid biopsies [25]. Liquid biopsies are meant to replace the existing biopsies obtained from the tumor mass itself, biopsies that are penetrate, with high cost, cause pain and clinical complications and contain a heterogeneous population of tumor as well as other non-cancer cells, thereby may be miss-interpreted [5]. Of the main two approaches: circulating tumor cells (CTCs) and circulating DNA, we focused on the former.

The isolation of CTCs is evolving as a promising approach for genetic analysis and therapeutic decisions in cancer and as a possible tool for detection of tumors. The available methods for isolation of CTCs are specific for various types of cancer and therefore the hunting of these CTCs is done by binding to a specific antibody which serves as a cancer marker, which varies among the different malignant cells as well as lacks in others. Herein we describe a somewhat novel approach in which we use the peptide of telomerase, the hallmark of cancer cells, as a bait to trap all cancer cell types based on the definition of the hTERT as a pan-cancer marker [26]. This definition stems from the uniqueness of telomerase reverse transcriptase—its presence in nearly all types of cancer cells (>90%) and its absence from most somatic cells. The approach of “anti-personalized medicine” has many advantages as it may overcome the above-mentioned pitfalls of the current methodology.

Telomerase containing cells present on their membrane telomerase driven peptides. Our strategy is based on one of them, termed GV1001, a 16 amino acid peptide (EARPALLTSRLRFIPK) which was used to construct an antibody for the identification of cancer cells expressing telomerase [21]. Our constructed antibody can specifically bind numerous cancer cell lines but not normal fibroblasts (Figure 1), albeit with different affinities. These differences may stem from the differential dependencies of the cells in telomerase activity. Along these lines, we have previously shown that T-cell leukemia (Jurkat) cells presented the highest telomerase activity compared to other cancer cell lines as well [27]. We have also analyzed the in vitro binding of our α-hTERT mAB by immunofluorescence and similarly, our antibody specifically bound to colon cancer cell lines but not to peripheral mononuclear cells obtained from a healthy donor (Figure 1B).

Since the α-hTERT antibody is present specifically on cancer cells membrane, it was shown to be successfully used as a cancer related antigen for developing anti-cancer vaccination in several studies to treat advanced pancreatic cancer, non-small cell lung cancer, melanoma, and other cancers [28,29,30,31,32].

Next, we showed that CLL cells obtained from patients differing in their CLL cells concentrations, bound our antibody. When we labelled the cells firstly with our antibody and only then with αCD19 and αCD5 as CLL markers, we got similar results, thus emphasizing the capability of the antibody to recognize and bind CLL cells (Figure 2, Figure 3 and Figure 4 and Figure 6). More importantly, this α-hTERT mAB bound also to cell suspension obtained from patients with ovarian tumor and breast cancer, PBMCs from Merkel carcinoma and Lung cancers, pointing to its potential usage in capturing CTCs (Figure 2 and Appendix A). Specifically, our antibody did not bind to healthy lymphocytes (Figure 5).

Our study has several limitations. First, hTERT is also expressed in actively dividing normal somatic cells, including small intestine, colon, lymph nodes and also hematopoietic stem cells, although to a much lesser extent compared to malignant cells [33,34]. In the current study we did not test our antibody against these specific cells. Hopefully in future studies the difference between the hTERT expression in these somatic versus neoplastic cells will be reflected by the differential binding of the antibody to these cells.

Second, the specificity of our mAb was evaluated by a parallel assessment of the binding of our mAB to ex vivo samples obtained from oncological patients and from healthy volunteers. To calculate the specificity and sensitivity of our antibody, we currently collect more samples in order to analyze these parameters by a Receiver Operating Characteristic (ROC) curve in the coming future. Third, we did not report the efficacy of our antibody, since only in CLL cells and not in the cells originated from the solid tumors we could compare the extent of the hTERT antibody binding to the presence of CLL markers on the neoplastic cells. All in all, we have characterized an anti-telomerase specific mAb which was shown to be specific to telomerase positive cells, both in vitro and ex vivo (Figure 7*)*. The fact that the extent of the antibody binding differed in various cell lines may stem from the different expression of the enzyme in these cancer cells. We have subsequently sequenced our specific antibody and provide herein the full sequence of both heavy and light chains of it.

Finally, to improve the specificity of the α-hTERT mAb we have cleaved it to F(ab’)2 fragment. As expected, the cleaved F(ab’)2 fragment bound cancer cells specifically ex vivo (Figure 6), in line with other studies reporting a similar improvement [35,36].

These results show that our anti-hTERT antibody-based system is able to identify and isolate circulating tumor cells of various origin.

One major pitfall in our study might have been that the antibody would not identify all cancer cells expressing telomerase and therefore the number of isolated CTCs will not be maximal; This point already proven not to be a problem, as our antibody is highly sensitive and specific for cancer cells. However, the advantages in the development of this “CTC trapping” anti “personalized medicine” based approach are clear: it is an easy cheap and noninvasive one.

Hopefully the approach that we developed will be thoroughly explored in future studies which may result in the development of a more beneficial method to trap CTCs for the benefit of patients with cancer.

## 4. Materials and Methods

### 4.1. Cell Culture

Cells were grown under standard growth conditions. The following cells were used: Jurkat (T cell leukemia), MCF-7 (breast adenocarcinoma), U266 (multiple myeloma), MEL (melanoma), JeKo-1 (mantle cell lymphoma), A549 (lung carcinoma). Human mononuclear cells (MNCs) from healthy volunteers and human fibroblasts served as a negative control. All cell lines were originated from the American Type Culture Collection (ATCC). U-266, Jurkat, JeKo-1 and MEL were cultured in RPMI-1640, A549 and MCF-7 in DMEM supplemented with 10–20% fetal bovine serum (FBS), 100 units/mL L-glutamine and 1% penicillin/streptomycin/nystatin (Biological Industries Beit Haemek, Israel) at 37 °C with 5% CO_2_. Jurkat and MEL cell media were supplemented with 10% FBS, U-266 was grown in the presence of 15% FBS and JeKo-1 with 20% FBS. Fibroblasts were propagated in DMEM with 20% FBS supplemented with L-glutamine and penicillin/streptomycin/nystatin.

Patients’ samples were obtained upon consent via protocol # 0038-12 approved by the IRB of Rabin Medical Center.

### 4.2. Isolation of Mononuclear Cells

Peripheral blood samples were collected and stored at room temperature and processed within 24 h of the collection. All participants signed an informed consent. Peripheral blood mononuclear cells (PBMCs) were isolated from the blood samples using standard protocol with Lymphoprep (Stem Cell Technologies, Cambridge, MA, USA).

### 4.3. Antibody Production

The hTERT peptide GV1001, EARPALLTSRLRFIPK, was synthesized by Genemed Synthesis (San Antonio, TX, USA). Hybridomas were produced by ProteoGenix (Schiltigheim, France). The Screening of the various antibodies containing supernatants was performed by ELISA and Flow Cytometry. The antibodies and isotype controls for flow cytometry were purchased from BioXCell Company (West Lebanon, NH, USA).

### 4.4. F(ab’)2 Production

F(ab’)2 fragments of α-hTERT mAb were prepared according to the protocol provided by Pierce mouse IgG1 Fab and F(ab’)2 preparation kit (ThermoScientific, MA, USA). The fragments were generated by using ficin in the presence of cysteine and analyzed by polyacrylamide gel electrophoresis (PAGE).

### 4.5. Enzyme-Linked Immunosorbent Assay (ELISA)

The specificity of monoclonal antibodies (mAbs) to the hTERT peptide was assessed by sandwich ELISA, using a standard protocol, wherein the tested mAbs binds to the hTERT peptide and not to the carrier protein (KLH) immobilized to the assay plate. Unbound antibodies were removed by washing with PBS. Subsequently, the bounded antibodies were detected by horse radish peroxidase (HRP)-conjugated anti-mouse IgG antibody and a 3,3′,5,5′-tetramethylbenzidine (TMB) substrate. The optical density was measured at 450 nm by microplate reader (Synergy HT, BioTek).

### 4.6. Flow Cytometry and Sample Preparation

The binding of the α-hTERT antibodies to a panel of cancer cells from patients was assessed by flow cytometry analysis. Blood, tumor and ascites samples were first stained with primary α-hTERT peptide mAb or isotype control, then incubated with FITC or AF647-conjugated α-mouse secondary Ab (Jackson, ME, USA) for the detection of hTERT positive cells.

Chronic lymphocytic leukemia (CLL) cells were marked as CD19+/CD5+ (PE-CD19 and PerCP-CD5, BioLegend, CA, USA); ovarian cancer cells were detected by following the marker CCCR5. Antibodies were purchased from ENCO, Jackson, Sigma and Abcam, CA, GB).

All cells were prepared for the flow cytometry analysis according to a routinely used procedure. Cell lines: floating cells collected from the growth media by centrifugation, adherent cells were trypsinized. Peripheral blood mononuclear cells (PBMCs) obtained from whole blood were isolated by Lymphoprep (Stem Cell Technologies, Cambridge, MA, USA).

The flow cytometry data was acquired using a BD FACS Calibur flow cytometer (BD BioSciences, San Jose, CA, USA), acquisition and analysis were performed using Cell Quest Pro software (BD Biosciences, USA).

PBMCs were centrifuged, the pellet was resuspended in PBS +2% FBS and equally split into tubes. Cells were subsequently stained with α-hTERT or isotype mAb, washed with Phosphate Buffer Saline (PBS) and incubated with fluorescently labeled secondary anti-mouse Ab (Sigma, MI, USA). Following washing cells were stained with tumor markers antibodies and re-suspended in PBS for acquisition on a flow cytometer. The same setup and compensation were used for all CLL samples. Cell debris and aggregates were excluded on forward scatter vs. side scatter. Data is presented by dot plots, histograms and graphs.

### 4.7. Protein Sequencing: (Edman Degradation)

The sequencing of the light chain of the antibody was performed using Edman degradation, in gel protein digestion and liquid chromatography-mass spectrophotometry (LC-MS) conducted by the Mass Spectrometry (MS) biological services unit of the Weizmann Institute of Science, Rehovot.

The heavy chain of the purified antibody was cleaved by trypsinization prior to LC-MS analysis.

### 4.8. Nucleic Acid Sequencing

The nucleotide sequence was obtained by using degenerated primers based on predicted sequences designed at the Tel Aviv University. The Kapa HiFi HotStart PCR (Thermo Fisher MA, USA) was used for the PCR reaction and the procedure was performed according to routinely used protocol. The nucleic acid sequencing was performed by the Instrumentation and Service Center at Tel Aviv University.

### 4.9. Immunofluorescence

Immunofluorescence analysis was conducted as described previously [26]. Basically, cells were fixed with 2% paraformaldehyde in PBS, permeabilized by 0.1% Triton X-100 in PBS and blocked with 2% bovine serum albumin in PBS overnight at 4 °C. Thereafter cells were washed with blocking buffer and incubated with our α-hTERT mAb or α-CD45 ab (Abcam, Cambridge, UK) for 16 h followed by incubation with labeled secondary antibodies (AF594 α mouse or AF488 α -rabbit respectively). To visualize the cells, we dropped them on glass slides mounted with Vectashield mounting medium with DAPI (Vector Laboratories, Newark, CA, USA) to counterstain nuclei. Images were inspected by using a Zeiss LSM510 Meta laser scanning confocal microscope (Carl Zeiss, Oberkochen, Germany) equipped with a Zeiss Axiovert 200 M.

## Figures and Tables

**Figure 1 ijms-23-12872-f001:**
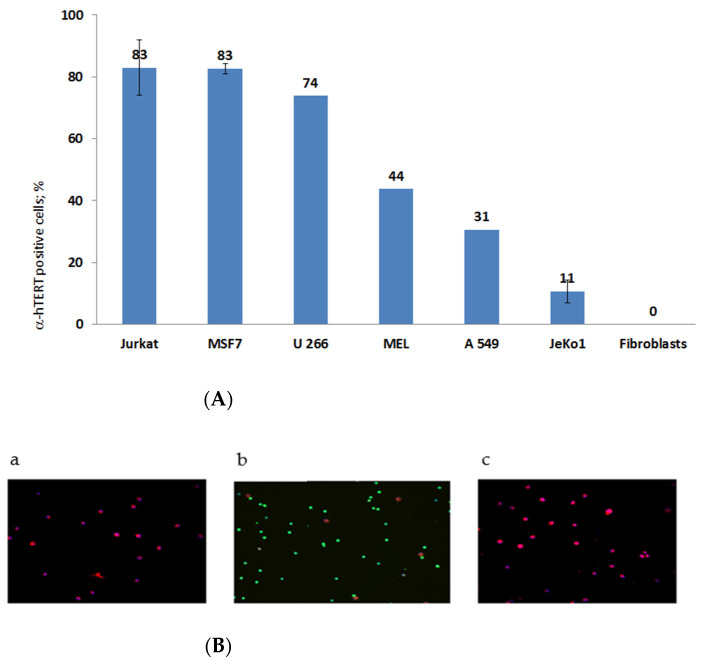
(**A**). flow cytometry analysis. Cells were grown in standard conditions according to the ATCC recommendations. The binding of the α-hTERT mAb to the cells was detected by flow cytometry using a secondary labeled antibody as described in the Materials and Methods section. (The number of repeats for each cell line is: Jurkat—8; human Fibroblasts—3; MSF7 and JeKo1-2; U266, MEL, A549-1). (**B**). Immunofluorescence analysis. Adenocarcinoma of the colon, SW480 cells (**a**); PBMC from a healthy donor (**b**) and colorectal carcinoma cells, HCT116 (**c**) were labeled with the α-hTERT mAb and CD45. Anti-mouse AF594 (Red) and anti-rabbit AF488 (Green) were used as a secondary antibody and DAPI (blue) was used as a nuclear counterstain.

**Figure 2 ijms-23-12872-f002:**
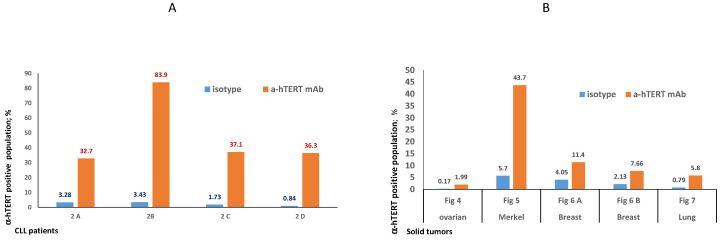
Ex vivo binding of the α-hTERT mAb to cancer cells obtained from patients with CLL (**A**) and from patients with solid tumors (**B**). (**A**). Binding to PBMCs from patients with CLL: 2A— a CLL patient with low amount of CLL cells (<50%), 2B—a CLL patient with high amount of CLL cells (>80%); 2C—a newly diagnosed CLL patient; 2D—a CLL patient after treatment. (**B**). Binding to circulating neoplastic cells and tumor cells suspension from patients with solid tumors as depicted in the X axis. The relevant dot plots are provided in the corresponding Appendix A.

**Figure 3 ijms-23-12872-f003:**
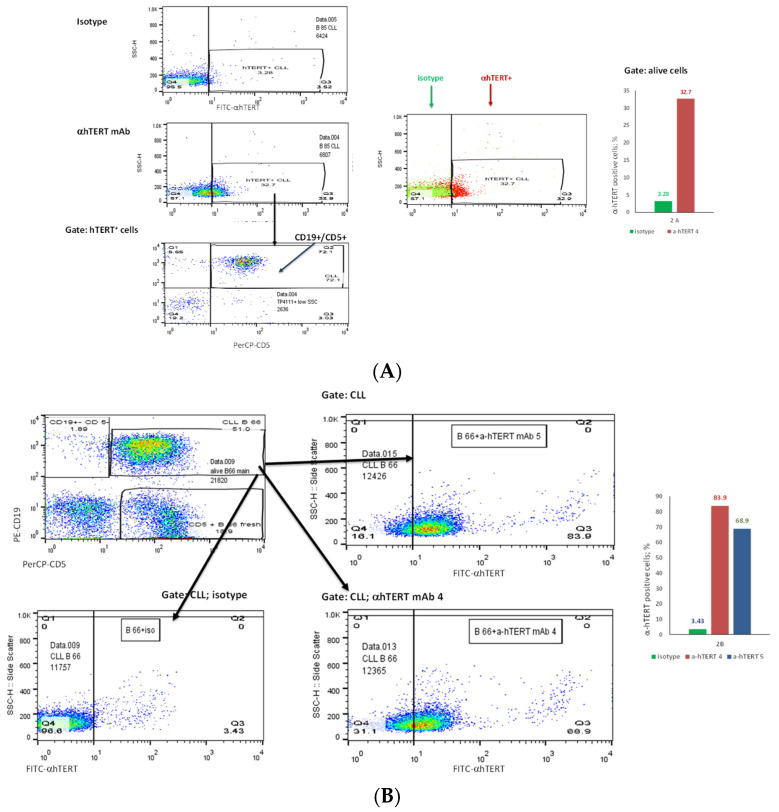
The binding of the α-hTERT mAb to PBMCs of CLL patients. The extent of the binding is shown on the right in each panel. (**A**). Most of α-hTERT positive cells are also CLL positive (CD19+/CD5+). PBMCs of CLL patient first were incubated with α-hTERT mAb or its isotype, then washed and stained with FITC-α-mouse secondary antibody followed by CLL markers PE-CD19/PerCP-CD5 staining. All α-hTERT positive cells were gated and analyzed for CD19/CD5. Overlay flow cytometry dot plot: the isotype is marked with green, the α-hTERT positive cells—in red. (**B**). CLL (CD19+/CD5+) Cells obtained from Patients with CLL are α-hTERT Positive. CLL cells from patient were gated according to the phenotype markers CD19/CD5, incubated with the α-hTERT mAb or its isotype IgG and stained with FITC-a-mouse secondary Ab. Flow cytometry dot plots show the binding α-hTERT mAb to CD19+/CD5+ cells and almost no binding of the isotype IgG. (**C**). *Ex vivo* binding of the α-hTERT mAb to CD19+/CD5+ cells from a newly diagnosed patient with CLL. Flow cytometry analysis: incubation with α-hTERT mAb or its isotype, staining with FITC-α-mouse secondary antibody followed by CLL markers PE-CD19/PerCP-CD5. (**D**). *Ex vivo* binding of the α-hTERT mAb to CD19+/CD5+ Cells from Patient with CLL during treatment. Flow cytometry analysis was conducted as above.

**Figure 4 ijms-23-12872-f004:**
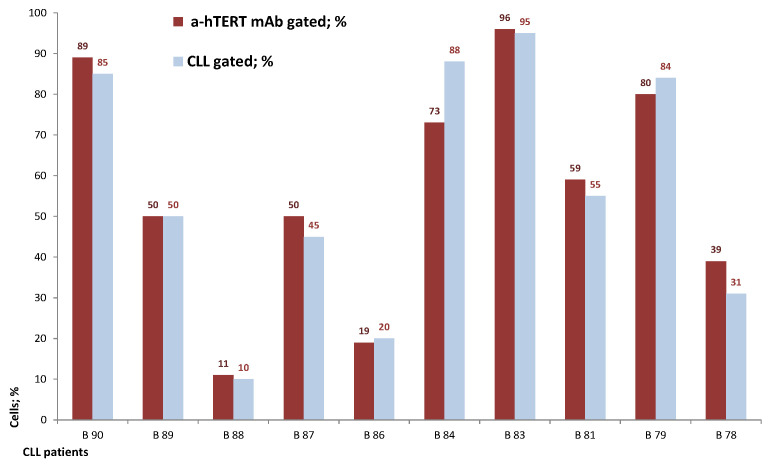
The percentage of the α-hTERT mAb positive cells corresponds to the percentage of the ex vivo obtained neoplastic (CLL: CD19+/CD5+) cells. A similar analysis as described in Figure 3 was conducted.

**Figure 5 ijms-23-12872-f005:**
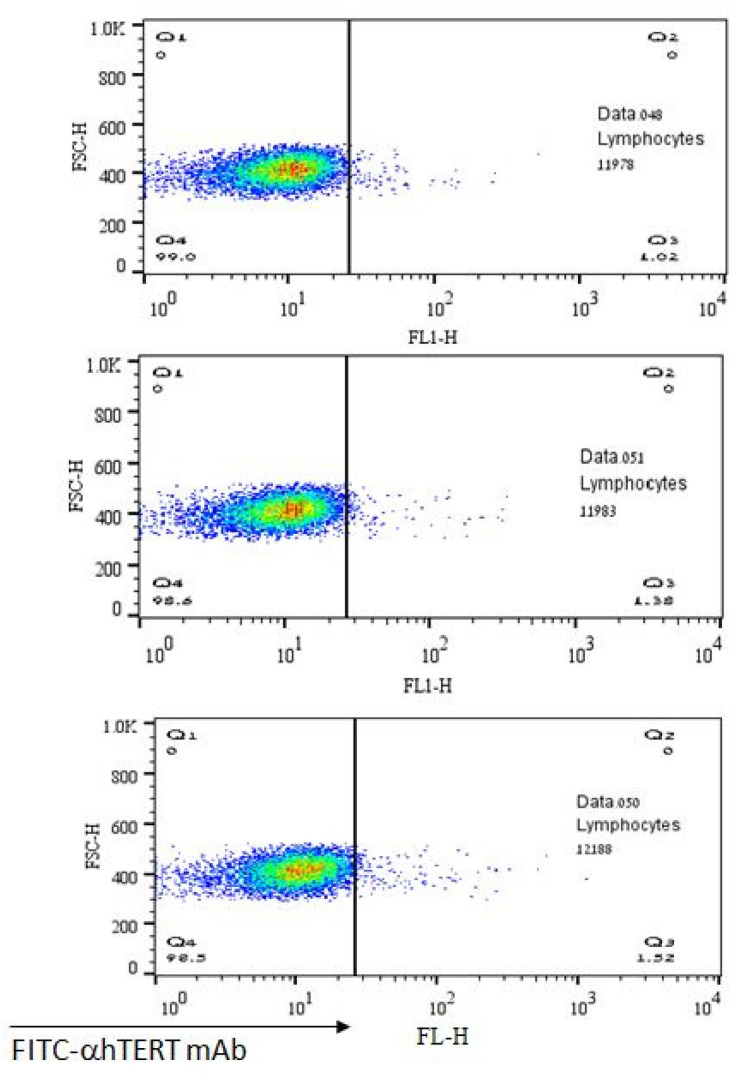
No binding of the α-hTERT mAb to lymphocytes of healthy volunteers. The α-hTERT mAb was added to PBMCs of healthy donors, stained with FITC-α-mouse secondary antibody as described in the Materials & Methods and analyzed by flow cytometry.

**Figure 6 ijms-23-12872-f006:**
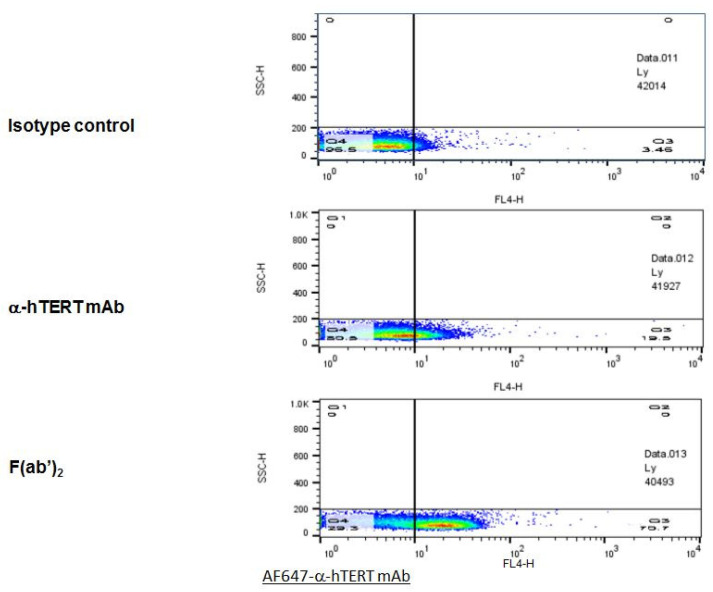
The F(ab’)2 fragment of the α-hTERT mAb effectively binds to PBMCs of CLL Patients. A Representative dot plot is shown. PBMCs of CLL patient were incubated with the α-hTERT whole mAb, or it’s F(ab’)2 fragment, or the appropriate isotype control. The binding was visualized by AF647-a-mouse secondary Ab.

**Figure 7 ijms-23-12872-f007:**
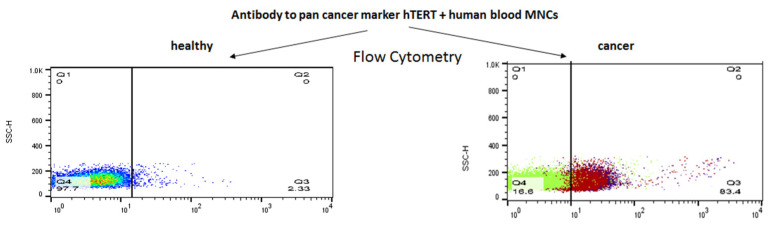
A summary of our α-hTERT mAb specificity to cancer cells.

## Data Availability

Not applicable.

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
