# Peer review of "Identification of Cancer Cells in the Human Body by Anti-Telomerase Peptide Antibody: Towards the Isolation of Circulating Tumor Cells"

_ijms, 2022, doi:10.3390/ijms232112872_

Round 1

Reviewer 1 Report

The authors described development of a monoclonal antibody against hTERT peptide (a-hTERT mAb) presented on the surface of cancer cells. This a-hTERT mAb has been previously reported. The authors attempt to test its possible applications as a pan-cancer marker for diagnostic, theragnostic and therapeutic possibilities. This is an interesting study. The results showed that this antibody is capable of detecting various types of cancer cells both in vitro and ex vivo from tumors of patients. If the work includes a test showing that this antibody is good enough or have potential feasibility for diagnostic with certain specificity and efficiency would be great.

Author Response

The authors described development of a monoclonal antibody against hTERT peptide (a-hTERT mAb) presented on the surface of cancer cells. This a-hTERT mAb has been previously reported. The authors attempt to test its possible applications as a pan-cancer marker for diagnostic, theragnostic and therapeutic possibilities. This is an interesting study. The results showed that this antibody is capable of detecting various types of cancer cells both in vitro and ex vivo from tumors of patients.

If the work includes a test showing that this antibody is good enough or have potential feasibility for diagnostic with certain specificity and efficiency would be great.

We deeply thank the reviewer for reading thoroughly our manuscript.

Just a minor correction- "This antibody has never been reported", probably the reviewer meant this peptide" (since our antibody is novel)..

The sensitivity and specificity test that is suggested by the reviewer is a good idea. However, given our current results we cannot conduct the ROC curve that is needed to calculate specificity and sensitivity. As more results will be collected this point will be addressed for sure. We have added a paragraph in the discussion as a limitation of the study referring also to the efficacy of the antibody (lines 350-363).

Reviewer 2 Report

  1. hTERT also expressed in actively dividing normal human cells, including small intestine, colon, and lymph node or more important, hematopoietic stem cells.
  2. On line 52, 109 should be upper cased
  3. Is there any HLA type restriction on the anti-hTERT antibody recognition?

Author Response

  1. hTERT also expressed in actively dividing normal human cells, including small intestine, colon, and lymph node or more important, hematopoietic stem cells.
  2. On line 52, 109 should be upper cased
  3. Is there any HLA type restriction on the anti-hTERT antibody recognition?

We deeply thank the reviewer for his critical reading of our manuscript. With regards to the comments:

  1. hTERT is indeed expressed in dividing normal cells, however, to a lesser extent than tumor cells. We added a paragraph regarding this point in the discussion as a limitation of the current study (lines 350-363).
  2. This has now been corrected, thanks! (line 52- Red Blood Cells, Line 109- a This was mended also throughout the text. I hope that this was your intention…
  3. We are not aware of any HLA type restriction on the anti-hTERT antibody recognition (added in the discussion as well).

Round 2

Reviewer 1 Report

The authors explained my concern and I have no further question.